# Head-Mounted Projector for Manual Precision Tasks: Performance Assessment

**DOI:** 10.3390/s23073494

**Published:** 2023-03-27

**Authors:** Virginia Mamone, Vincenzo Ferrari, Renzo D’Amato, Sara Condino, Nadia Cattari, Fabrizio Cutolo

**Affiliations:** 1EndoCAS Center for Computer-Assisted Surgery, University of Pisa, 56124 Pisa, Italysara.condino@endocas.unipi.it (S.C.); nadia.cattari@endocas.unipi.it (N.C.); fabrizio.cutolo@unipi.it (F.C.); 2Azienda Ospedaliero Universitaria Pisana, 56126 Pisa, Italy; 3Information Engineering Department, University of Pisa, 56126 Pisa, Italy

**Keywords:** spatial augmented reality, optical tracking, head-mounted display, head-mounted projector, motion-to-photon latency

## Abstract

The growing interest in augmented reality applications has led to an in-depth look at the performance of head-mounted displays and their testing in numerous domains. Other devices for augmenting the real world with virtual information are presented less frequently and usually focus on the description of the device rather than on its performance analysis. This is the case of projected augmented reality, which, compared to head-worn AR displays, offers the advantages of being simultaneously accessible by multiple users whilst preserving user awareness of the environment and feeling of immersion. This work provides a general evaluation of a custom-made head-mounted projector for the aid of precision manual tasks through an experimental protocol designed for investigating spatial and temporal registration and their combination. The results of the tests show that the accuracy (0.6±0.1 mm of spatial registration error) and motion-to-photon latency (113±12 ms) make the proposed solution suitable for guiding precision tasks.

## 1. Introduction

In recent years, the potential of Augmented Reality (AR) technology to enhance the real world with auxiliary virtual information has become increasingly prominent. To this end, Head Mounted Displays (HMDs) are by far the most studied devices. Their application has been proposed in various fields ranging from industrial quality control and assembly [1,2], education [3], military [4,5], and surgery [6,7] to entertainment [8]. In the literature, the characteristics of HMDs have been extensively studied and a few drawbacks have emerged such as the difficulty in achieving a feeling of immersion, the loss of spatial awareness [9], and the development of motion sickness [10]. These drawbacks contribute to an increased cognitive load resulting from well-known technological pitfalls typical of commercial and/or laboratory prototypes of HMDs. Among the technological drawbacks that limit the success of AR HMDs in the consumer market, there is the presence of a narrow display field of view, the discrete and low display resolution, the low frame rate, the perceivable motion-to-photon latency, and the perceptual issues such as the vergence-accommodation conflict and the real-to-virtual focus rivalry. These limitations strongly impact the perceptual experience and result in maladaptations, such as cybersickness [11], other than producing safety concerns. In addition, these limits greatly hinder the widespread adoption of AR HMDs to support the accurate and efficient execution of highly challenging manual activities that must be carried out in the peripersonal space (i.e., for surgical guidance) [12], with alternative AR solutions that have been proposed to overcome them.

Projected AR, also called spatial AR, lends itself to preserving the user’s awareness of the environment and the feeling of immersion since it does not require any optical interface interposed between the user’s eyes and the surrounding world. In projected AR, physical and digital information are mixed directly into the real world via visible light projectors. As a result, a single projected AR device can be simultaneously experienced by multiple users. However, such devices are mostly static, as they are installed in a fixed position, hence they inevitably constrain the user’s freedom of movement. With the development of increasingly smaller and lighter projectors, new solutions have been proposed to overcome this problem. Nazarova et al., for instance, mounted a tracking and projection module on a collaborative robot to provide immersive and intuitive robot control to the user [13]. Hartmann’s team proposed an actuated Head-Mounted Projector (HMP) to increase the user’s field of view and foster collaboration and communication by providing a shared experience with other people [14]. MoSART, instead, identifies itself as an alternative to classic HMDs through the development of an “all-in-one” headset that includes projection and tracking systems to provide AR for 3D Interaction with tangible objects [15]. The critical limitation of these systems is widely identified as the motion-to-photon (MTP) latency.

The term MTP latency was originally used to denote the average time between the instant when a motion occurs and the instant when the corresponding rendering event gets displayed on the display (i.e., in front of the user’s eyes for an HMD). MTP latency has been extensively studied for HMDs and literature results reveal their major impact on the system’s usability [10] since high latency can lead to substantial dynamic registration errors between the real world and that virtual content that heavily compromise the immersiveness of the experience, and it can also generate motion sickness and user discomfort [16]. A study by Jerald and Whitton showed that a 5 ms latency is completely imperceptible to a user in HMDs [17]. The Microsoft HoloLens comes very close to this value as they compensate for the slow optical tracking constrained by tracking the camera’s image acquisition time and processing time by exploiting tracking data from IMU sensors mounted on the headset coupled with motion-prediction 2D warping mechanisms. The latency of Meta 2 is about 60 ms [15], whereas that of HoloLens 2 is just 9 ms [18]. The stability of the virtual content, on the other hand, comes in terms of jitter. Jitter can be extremely disturbing for a human observer and can impair the feeling of immersion by breaking the illusion of a mixture between the virtual and real world. Jitter for Meta 2 and HoloLens is 1.7 mm and 0.2 mm at rest, respectively. A jitter of 1.7 mm has been identified as the upper limit for HMDs [19]. The same data are not available for HMPs proposed in the literature, although latency and jitter are recognized as critical aspects in the use of such devices. In [14], only the projector latency of 116 ms is given, with no mention of latency due to tracking or rendering that necessarily worsens the final value. MoSART system, on the other hand, claims a latency of 60 ms and jitter of 0.08 mm (MSE) but the same paper does not detail how these values were measured.

In this work, we propose an HMP built by assembling commercial devices, and we provide a detailed description of the assessment of its spatial and temporal registration together with their influence on the usability of the worn system [20]. The spatial registration accuracy depends on the geometry behind the AR implementation process, including calibration, tracking mechanism, and modeling of the target structures. Temporal registration comes into play when the user or the tracked object moves and it corresponds to synchronized motion between real and virtual structures over time. MTP latency is a measure of temporal registration. Specifically, we here present a protocol for:Measure the temporal registration error of the HMP in terms of MTP latency.Measure the static spatial registration error in terms of accuracy by evaluating the distance between the actual position of the virtual content and the desired position for perfect alignment in static conditions.Measure how these two errors combine when a user wears the device and uses it to augment a tracked structure in real time.

All measurements were taken within the working area accessible to a user who intends to perform manual tasks while wearing the HMP. This area is between 350 mm and 600 mm from the user’s head, with an average value of 475 mm (i.e., the area within the user’s peripersonal space).

## 2. Materials and Methods

First, we present the hardware specifics of the HMP together with a brief explanation of the software framework that implements the tracking and the rendering. Then, the experimental protocol employed to evaluate the spatial and temporal registration accuracy and the combined effect of MTP latency and virtual-to-real static registration is described during a user test.

### 2.1. The Custom-Made Head-Mounted Projector

The custom-made HMP includes a Leopard Imaging OV580 stereo camera rig and an Anybeam laser mini-projector. The adopted camera configuration is 2560 × 720 @ 60 fps, which corresponds to an angular resolution of about 3.6 arcmin/px. The AnyBeam projector offers the advantage of no need for focus adjustment, as this is ensured by the MEMS laser technology. Projector resolution and brightness are, respectively, 720p @ 60 fps and 150 lumens. The stereo rig and the projector are calibrated to estimate both their intrinsic and extrinsic parameters [21,22]. The stability of the reciprocal pose of the components is ensured by a 3D-printed support modeled in Creo Parametric 3.0. The same support features an elastic band that allows the device to be worn on the head for an overall weight of about 300 g. The HMP is plugged into the computing unit where the software framework runs, consisting of a laptop with Intel Core i7-9750H CPU@ 2.60 GHz, 6 cores, 12 GB RAM, and GeForce RTX 2060 GPU by Nvidia.

### 2.2. Augmented Reality Software Framework

The software framework that controls the HMP was thoroughly described in [23]. Here, we recall its main features:The software, originally conceived for the deployment of AR applications on video see-through (VST) and optical see-through (OST) display interfaces, is capable also of supporting the deployment of AR applications on different typologies of AR projectors. The software features a non-distributed architecture, which makes it also compatible with embedded computing units.The software framework is based on Compute Unified Device Architecture (CUDA by Nvidia) to leverage parallel computing over the multi-core GPU card.The software supports in situ visualization of 3D structures, thanks to the employment of the open-source library VTK for 3D computer graphics, modeling, and volume rendering.The software framework is highly configurable in terms of rendering and tracking capabilities.The software features a robust inside-out video-based tracking algorithm based on OpenCV API 3.4.1,

The software framework is able to support in situ visualization of medical imaging data; the key function of the software, under projector AR modality, is to process the images grabbed by the stereo pair of RGB cameras. The grabbed frames of the real scene are processed to perform marker-based optical tracking, which requires the identification of the 3D position of the markers both in the target reference frame and camera reference frame. The projected AR device needs a dedicated calibration between the world-facing tracking cameras and the projector. For this purpose, the projector is considered as an inverse camera according to the procedure exposed in [24]. Such calibration routine outputs the intrinsic projection parameters of the projector and the relative pose between the left camera and the projector itself.

### 2.3. Experimental Protocol

The experimental protocol relies on an external device for the evaluation of spatial and temporal registration accuracy. We selected a GoPro Hero 5 Black camera that features either a 4000 × 3000 photo resolution or a frame rate of up to 240 fps for videos at a 1280 × 720 resolution. As shown in Table 1, we adopted the highest resolution configuration for the static spatial registration tests and the highest frame rate for the dynamic temporal registration tests. For both configurations, we calibrated the GoPro Hero 5 and derived the average angular resolution (the amplitude of the visual angle covered by each pixel in arcmin) to enable valid comparison of the results. Table 1 also reports the working depth in order to obtain an estimate of the accuracy in physical distances (i.e., in mm).

#### 2.3.1. Spatial Registration

We evaluated the AR spatial registration accuracy as the on-image distance between the location of a real point and its virtual counterpart (i.e., we measured the registration error in terms of target visualization error). For this purpose, we adopted a spatial registration evaluation tool consisting of three spherical markers fixed on a plane featuring two elevated points. The tool was modeled using Creo Parametric 3.0 and printed by rapid prototyping with a 600 dpi X-Y resolution, 1600 dpi Z resolution, and 0.1 mm precision. The two target points were dyed in red to be easily identified in the images as they are the reference points for AR alignment error assessment. The experimental set-up is shown in Figure 1a along with the spatial registration evaluation tool. The HMP and the GoPro Hero 5 were mounted on a holder that allows their distance from the working plane to be varied. In this way, we evaluated the registration error over the entire range of working distances between 350 mm and 600 mm (with a 25 mm step), a range of distances compatible with manual tasks. For each distance, we used GoPro setting 1 to capture five images of the evaluation tool. The acquired sets of images were automatically processed in Matlab™(version R2020.a) to estimate the 2D Target Visualization Error, that is the offset, expressed in pixels, between the virtual and real objects in the image plane (i.e., the centroids of the virtual landmarks and those of the corresponding real ones). The images were processed in hue-saturation-value color space and the circular Hough transform was used to detect the position of the virtual and real landmarks: first, virtual landmarks were detected in each series of AR images; then, their real counterparts were detected in the real image. 3D-printed landmarks were searched within a region of interest centered on the position of the corresponding virtual landmarks. We varied the orientation of the evaluation tool for each image captured at the same distance in order to include tracking error variability in the analysis: the target tool was placed in five different random orientations with respect to the tracking camera to account for the effect of different camera-to-target perspectives on the overall registration error. Overall, we collected 50 images, and for each image, we evaluated the distance of the two projected points from their real counterparts for a total of 100 evaluations of the error. Figure 1b shows one of the frames captured for the study, with the real points and their projected counterparts highlighted after manual selection.

#### 2.3.2. Temporal Registration

For the evaluation of MTP latency, we chose to adopt the protocol proposed in [25]. Specifically, we fixed the HMP and the GoPro camera on the holder at a distance of 475 mm from the working plane, selected as the mean working distance for manual tasks. We placed the spatial registration evaluation tool used in previous tests on the working plane. We measured the latency as the time difference between the displacement of the evaluation tool caused by a physical event and the associated virtual event (i.e., the associated movement of the projected virtual object). The physical event is the shooting of a rubber band against the frame, whereas its associated virtual event corresponds to the update of the projection of the virtual point. We repeated the test 15 times and captured the scene with the external camera switched to setting 2. Next, we decomposed the output clips into their frames and visually analyzed them to identify the frame in which the tool moves after the physical event and the frame in which the virtual information consequently updates its location. The number of frames between these two instants was then translated in ms by means of the selected frame rate of 240 fps (setting 2).

#### 2.3.3. User Test

To evaluate how the spatial and temporal registration errors combine whilst the HMP is worn by a user, we adopted the dummy presented in [26]. The dummy consists of three spherical tracking markers attached to a portion of a 3D-printed skullcap derived from a human CT scan. A specific trajectory, traced through Creo Parametric software, was designed to test the system registration accuracy. This trajectory is projected over the dummy in the form of a virtual cutting line that simulates a craniotomy (i.e., the virtual trajectory). On the other hand, we traced a real cutting line on the dummy using a template to obtain a real reference on the correct position of the cutting line (i.e., the real trajectory). The template was designed to exactly match the phantom surface and it was equipped with attachment holes engaging into cylinders mounted on the phantoms. This features ensures a unique and stable mechanical placement of the template on the phantom surface. We asked a user to wear the HMP and employ it to provide an AR view of the cutting line. We captured the scene with the GoPro switched to setting 3, as shown in Figure 2a. We first decomposed the clip into its frames and converted them into Hue-Saturation-Value (HSV) scale. This allowed us to segment the images by adopting a threshold on the H and V channels to extract the real cutting line drawn in blue and the white-colored projected line. Then, the segmented areas were collapsed into a one-pixel line, corresponding to the center line of the real cutting line obtained through skeletonization. The overlay accuracy between the real and projected cutting lines was measured in terms of Hausdorff distance as done in [27], which is defined as the largest distance among all the distances of a point in one line to the closest point of the other line:(1)$$d_{H}=max\left\{\underset{x\in X}{\sup}\;\underset{y\in Y}{\inf}\;d\left(x,y\right),\;\underset{y\in Y}{\sup}\;\underset{x\in X}{\inf}\;d\left(x,y\right),\;\right\}$$

Figure 2b, shows the extracted real and virtual centrelines of the real and projected cutting lines, and it highlights the points identifying the Hausdorff distance. This process was repeated for each frame to estimate the total registration error. In addition, the stability of the projected line was evaluated in terms of jitter, which is the deviation from the mean of all the projected lines, and it is used to obtain an estimate of the stability of the virtual content. The latter information was derived by fitting the lines with a 7-order polynomial. The average line was then found by averaging the coefficients among all the captured frames.

**Figure 2 sensors-23-03494-f002:**
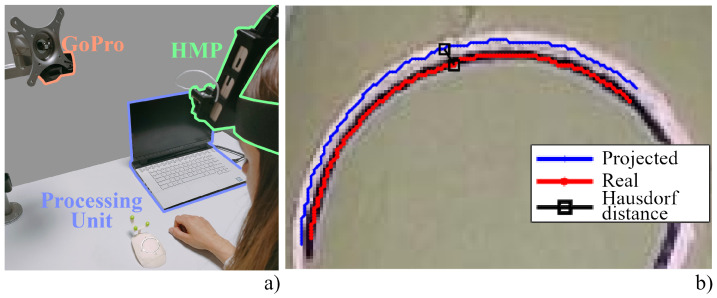
User test. (**a**) Set-up for the user test consisting of the Head-mounted Projector (green), the GoPro Hero 5 (orange), and the processing unit (blue). (**b**) Hausdorff distance between the projected center line (blue) and the real one (red). Points resulting from the Hausdorff algorithm are marked in black and the line connecting them is the Hausdorff distance between the two curves.

## 3. Results and Discussion

### 3.1. Tests Results

The spatial registration error within the working area was 11±2 px, which corresponds to 4.6±0.8 arcmin for setting 3 of the external camera. Given the working distance of 475 mm, we estimated a spatial registration error of about 0.6±0.1 mm (Table 2). These spatial registration errors belong to the ranges found in the recent literature for projected AR devices [24,28].

For the temporal registration, the number of frames elapsed between the motion of the evaluation tool and the update of the virtual information is 27±3 on average. Given the acquisition rate of 240 fps (setting 2 in Table 1), this is associated with an average MTP latency value of 113±12 ms (Table 3) This value is far lower than what is expected for [14], where 116 ms corresponds to the latency of the projector alone. On the other hand, the latency obtained in this work amounts is almost twice that reported for the MoSART system [15]. However, in both works, the authors provide no details on the latency evaluation, nor on the tracking or rendering times, thus no insights can be gained into the source of these differences.

We recorded a 25 s clip during the user tests which corresponds to about 6000 frames. Each frame was processed to estimate the centreline of the real and the projected cutting lines. Their Hausdorff distance is plotted in Figure 3a. The average error was found to be 5±1 px corresponding to 6.6±1.1 arcmin and 0.9±0.2 mm (Table 2). The registration error with the HMP worn by a user is higher than the error found for spatial registration alone. This may be due to small variations in the user’s head pose, which combined with the temporal registration error, result in a higher misalignment between real and virtual trajectories. However, the registration error falls within the acceptable range for performing precision manual tasks such as surgery [29]. The jitter with respect to the mean trajectory is plotted in Figure 3b. The RMS of the jitter is 2 px, equal to about 2.5 arcmin and 0.4 mm. This value is close to the one reported for MoSART (0.08 MSE = 0.3 RMS).

### 3.2. Ergonomics Considerations for Future Development

In terms of perceived user comfort and ergonomics, weight distribution in head-worn displays and projectors is one of the main limiting factors that hinder the widespread adoption of mobile AR devices in the healthcare sector and specifically in image-guided neurosurgery [30,31]. By way of example, the authors of [31] state that, since much of the weight on the Microsoft HoloLens is placed on the user’s forehead and nose, this may force the user to progressively shift his/her gaze downwards over time. This gradual shift in posture could therefore cause neck strain and general discomfort. Similar results were also obtained in [32], a user study where several participants reported back and neck strain as they were forced to move the weight of their head and of the display away from their body. Therefore, and in accordance with the findings of Ito et al. [33], improving the comfort of head-worn displays requires specific considerations regarding weight balance. Therefore, if on the one hand our HMP strictly fulfills the ergonomics requirement in terms of total weight to be <350 g, additional work has to be carried out to improve weight distribution and ergonomics so that the HMP can be worn for a prolonged time.

## 4. Conclusions

We proposed a device for projected AR, which can be worn by the user to guide the execution of precision manual tasks while preserving mobility in the working area. Compared to HMDs, our HMP offers the advantages of being simultaneously accessible by multiple users, without altering their awareness of the environment and feeling of immersion. We evaluated the performance of our HMP in terms of spatial and temporal registration along with their mutual interaction in a user test. The latency of 113 ms is a positive result, especially considering that the HMP is not equipped with an IMU sensor, nor with predictive slam algorithms or fast-rendering engines such as the ones adopted in the HoloLens. Specifically, the HMP meets the specifications proposed in [34] for HMDs, stating that “latencies shall be not longer than 150 milliseconds”. Future work will be aimed at identifying any critical points to further minimize the latency and evaluate whether it impacts task performance, sickness and discomfort [10,16] even in the case of projected AR. The jitter of 0.4 mm lies within the recommended range for HMDs [19]. It is slightly larger than the value provided for MoSART and twice the value reported for HoloLens 2. Although MoSART offers better performance in terms of latency and jitter, its overall weight and footprint are significantly higher with respect to the device investigated in this study. Specifically, the weight of 1 kg is likely to compromise the usability of the HMP in various applications such in military and medical fields [30,35].

To the best of the authors’ knowledge, our evaluation of spatial and temporal registration, together with their mutual interaction in a user test constitutes the first overall assessment of an HMP for precision manual tasks. We hope that this will provide a reference for further investigation of this and other AR HMPs and HMDs contributing to providing a standard for their performance assessment.

## Figures and Tables

**Figure 1 sensors-23-03494-f001:**
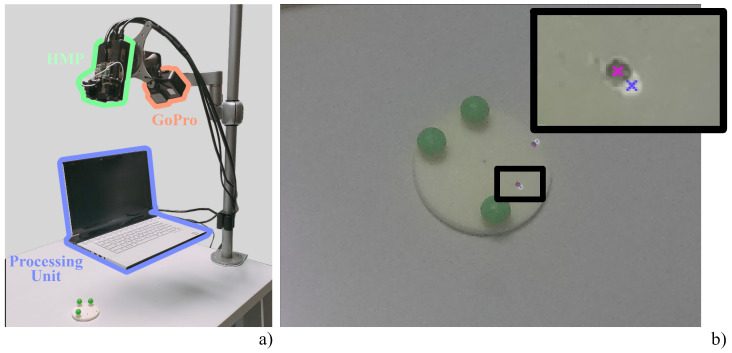
Spatial registration. (**a**) Set-up for evaluation of spatial registration consisting of the HMP (green) and the GoPro Hero 5 (orange), both placed on a holder for working depth selection, the processing unit (blue), and the spatial registration evaluation tool. (**b**) Zoom of an image captured by the GoPro Hero 5: pink crosses denote the real reference points, whereas blue crosses denote the projected points.

**Figure 3 sensors-23-03494-f003:**
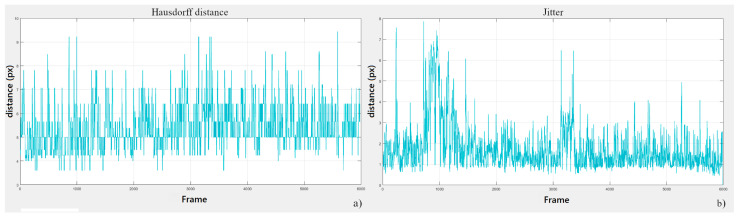
Results of the user test. (**a**) Hausdorff distances between the real trajectories and their corresponding projected trajectories. (**b**) Jitter of the projected trajectories.

**Table 1 sensors-23-03494-t001:** GoPro Hero 5 black: settings and specifications.

Setting	Test	Resolution	Frame-Rate	Angular Resolution	Working Depth
1	Spatial registration	3000 × 4000	-	0.40 arcmin/px	350–600 mm
2	Temporal registration	1280 × 720	240 fps	1.23 arcmin/px	475 mm
3	User test	1280 × 720	240 fps	1.23 arcmin/px	475 mm

**Table 2 sensors-23-03494-t002:** Spatial registration errors.

Setting	Test	On Image Registration Error (px)	Visual Angle Registration Error (arcmin)	Absolute Registration Error (mm)
1	Spatial registration	11±2	4.6±0.8	0.6±0.1
3	User test	5±1	6.6±1	0.9±0.2

**Table 3 sensors-23-03494-t003:** Temporal registration errors.

Setting	Test	Motion-to-Photon Latency (ms)
2	Temporal registration	113±12

## Data Availability

The data presented in this study are available upon request from the corresponding author.

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
