# Peer review of "Head-Mounted Projector for Manual Precision Tasks: Performance Assessment"

_sensors, 2023, doi:10.3390/s23073494_

Round 1

Reviewer 1 Report

The manuscript introduces a detailed description of a custom-made head-mounted projector (HMP)  for precision manual tasks and evaluates its performance through an experimental protocol. The specific focus on a custom-made head-mounted projector for precision manual tasks is relatively original.

The device was evaluated for its spatial and temporal registration in a user test, which is the first overall assessment of an HMP for precision manual tasks. The device achieved accuracy with a low latency of 113 ms, meeting the specifications proposed for HMDs. The device's jitter was slightly larger than some comparable devices but still within the recommended range for HMDs. The results of the tests show that the proposed solution is suitable for guiding precision tasks with accuracy even without advanced features like an IMU sensor or predictive slam algorithms.

It will be beneficial to discuss the impact of the custom-made HMP on user discomfort, especially during prolonged use. 

Author Response

Reply to Reviewer 1:

We thank the reviewer for his/her suggestion. We have integrated the discussion with a new subsection entitled “Ergonomics Considerations for Future Development” and containing observations pertaining the importance of weight and weight-balance in Head-mounted displays. The same requirements can be reasonably applied also to head-mounted projectors. This is the added part:

"In terms of perceived user comfort and ergonomics, weight distribution in head-worn displays and projectors is one of the main limiting factors that hinder the widespread adoption of mobile AR devices in the healthcare sector and specifically in image-guided neurosurgery [30,31].

By way of example, the authors of  [31] state that, since much of the weight on the Microsoft HoloLens is placed on the user’s forehead and nose, this may force the user to progressively shift his/her gaze downwards over time. This gradual shift in posture could therefore cause neck strain and general discomfort.

Similar results were also obtained in [32], a user study where several participants reported back and neck strain as they were forced to move the weight of their head and of the display away from their body.

Therefore, and in accordance with the findings of Ito et al. [33], improving the comfort of head-worn displays requires specific considerations regarding weight balance. Therefore, if on the one hand our HMP strictly fulfils the ergonomics requirement in terms of total weight to be be < 350 g, additional work has to be carried out to improve weight distribution and ergonomics so that the HMP can be worn for a prolonged time."

[30] D’Amato, R.; Cutolo, F.; Badiali, G.; Carbone, M.; Lu, H.; Hogenbirk, H.; Ferrari, V. Key Ergonomics Requirements and Possible Mechanical Solutions for Augmented Reality Head-Mounted Displays in Surgery. Multimodal Technol. Interact. 2022, 6, 15. https://doi.org/10.3390/mti6020015

[31] Nguyen, N.Q.; Cardinell, J.; Ramjist, J.M.; Androutsos, D.; Yang, V.X. Augmented reality and human factors regarding the neurosurgical operating room workflow. Optical Architectures for Displays and Sensing in Augmented, Virtual, and Mixed Reality (AR, VR, MR). SPIE, 2020, Vol. 11310, pp. 119–125.

[32] Lin, C.; Andersen, D.; Popescu, V.; Rojas-Muñoz, E.; Cabrera, M.E.; Mullis, B.; Zarzaur, B.; Anderson, K.; Marley, S.;Wachs, J. A First-Person Mentee Second-Person Mentor AR Interface for Surgical Telementoring. 2018 IEEE International Symposium on Mixed and Augmented Reality Adjunct (ISMAR-Adjunct), 2018, pp. 3–8. doi:10.1109/ISMAR-Adjunct.2018.00021.

[33] Ito, K.; Tada, M.; Ujike, H.; Hyodo, K. Effects of Weight and Balance of Head Mounted Display on Physical Load. Virtual, Augmented and Mixed Reality. Multimodal Interaction; Chen, J.Y.; Fragomeni, G., Eds.; Springer International Publishing: Cham, 2019; pp. 450–460.

Reviewer 2 Report

The paper evaluates the accuracy of a head mounted projector intended for providing guidance to precision manual tasks. That kind of information is scarcely available in literature so the paper could be an useful asset.

For some points more development information is needed:
1) User test with HMP worn by the user. Because the motion of the head creates a MTP error it is required to state how did the user behaved during tests? Did he keep its head static, was he mimicking an operation task that could have imposed a significant head movement?
2) Results and discussion, line 197-following. The test case Hausdorff distance is compared to spatial registration result. The larger error of the test case is related to the head mounted device that can be subjected to head motion so can combine spatial and TMP errors. But there is one more parameter that changed: resolution is reduced from 3000x4000 to 1280x720. How this affect spatial precision? An interesting comparison could be done between static spatial registration made with 3000x4000 resolution and 1280x720. Is there a significant difference?
3) Why frame-rate of GoPro is reduced from Temporal registration to User test? A higher frame rate should not give a better estimate of MTP?

Graphical issues:
In figure 3 the axes numbers are not visible. Please increase font size.
Results and discussion: for improving readability, please summarize the results in a table, structured like table 1.

Bibliography:
The user test is about a surgical application. In the state of the art only one review reference has been provided, dated 2019. It's possible that newer works were published in the period, like:
"Mixed Reality-Based Support for Total Hip Arthroplasty Assessment" https://link.springer.com/chapter/10.1007/978-3-031-15928-2_14
It could be beneficial to search for and cite newer studies HMDs for surgical applications.

Author Response

We are thankful for the interest shown in our work. We provide point-by-point answers to the comments and suggestions below.

  • The user was asked to keep her head still as much as possible during the test. However, and as mentioned in the text, “the registration error with the HMP worn by a user is higher than the error found for spatial registration alone. This may be due to small variations in the user’s head pose, which combined with the temporal registration error, result in a higher misalignment between real and virtual trajectories.”. We also agree that during a real operation the head movements would be amplified and result in a slightly higher registration error.
  • In measuring the registration error, we must consider the following considerations. Because the error in pixels depends on the hardware resolution and the target distance (i.e., device- and depth-dependent), the overlay accuracy is reported in mm (device-independent but depth-dependent), and in visual angles in arcmin (device and depth independent). The angular resolution @720p is lower (1.23 arcmin/px) compared to the angular resolution of the GoPro at 3000x4000 resolution (0.4 arcmin/px). On the other hand, the temporal and user test measurements require a higher frame rate.
  • We thank the reviewer for noticing this discrepancy (it was a typo). The frame rate for both the temporal registration and the user test is the same (i.e., 240 fps). As a matter of fact, we have processed 6000 frames for 25 s of camera acquisition (=25s x 240fps). We have now corrected the frame rate in the table for the user test.

Graphical issues:

We have increased the size of the axes numbers in Figure 3 and added two tables with the results of the tests as suggested by the reviewer.

Bibliography:

We have added the suggested reference in the text.

Reviewer 3 Report

the paper is interesting but it needs some adjustments:

1. lines 124 - 129. Why did you selected the GoPro? how did you calibrate it?=

2. lines 132-136. Please explain better how you calculated the distance between the location of the real point and its virtual counterpart. Then how are you certain of the precision of the evaluation tool 3D printed? and what is its accuracy?

3. lines 142-143. Why only five images?

4. line 144: "We varied the orientation of the tool for each image captured at the same distance in order to include tracking error variability in the analysis. " please explain better

5. 2.3.3 needs a deepen explanation of the process especially lines 163 - 169. Why all these settings were decided?

Author Response

We thank the reviewer for his/her comments. We provide point-by-point answers to the comments and suggestions below:

  1. We selected the GoPro camera because it actually offers a quite high frame-rate (240 fps) that is particularly useful for measuring the motion-to-photon latency. Since the GoPro camera was used as a means to measure the virtual-to-real misalignment and/or the MTP, we did not need any sort of camera calibration (i.e., we did not have to estimate the intrinsic camera parameters of the camera for these measures).
  2. As properly suggested by the reviewer, we have added a description of how the target visualization error was computed: “The acquired sets of images were automatically processed in Matlab (version R2020.a) to estimate the 2D Target Visualization Error, that is the offset, expressed in pixels, between the virtual and real objects in the image plane (i.e., the centroids of the virtual landmarks and those of the corresponding real ones). The images were processed in hue-saturation-value colour space and the circular Hough transform was used to detect the position of the virtual and real landmarks: first, virtual landmarks were detected in each series of AR images; then, their real counterparts were detected in the real image. 3D-printed landmarks were searched within a region of interest centred on the position of the corresponding virtual landmarks.”
  3. and 4) We have expanded the description of the methods for measuring the spatial registration error as follows: “we varied the orientation of the target tool for each image captured at the same distance in order to include tracking error variability in the analysis: the target tool was placed in five different random orientations with respect to the tracking camera to account for the effect of different camera-to-target perspectives on the overall registration error. Overall, we collected 50 images, and for each image, we evaluated the distance of the two projected points from their real counterparts for a total of 100 evaluations of the error. We considered 50 possible tool orientations and placement as sufficient to have an estimation of the spatial registration error."

    5)  At lines 163-169, we have reworded and expanded the section as follows: “To evaluate how the spatial and temporal registration errors combine whilst the HMP is worn by a user, we adopted the dummy presented in [25]. The dummy consists of three spherical tracking markers attached to a portion of a 3D-printed skullcap derived from a human CT scan. A specific trajectory, traced through Creo Parametric software, was designed to test the system registration accuracy. This trajectory is projected over the dummy in the form of a virtual cutting line that simulates a craniotomy (i.e., the virtual trajectory). On the other hand, we traced a real cutting line on the dummy using a template to obtain a real reference on the correct position of the cutting line (i.e., the real trajectory). Ad hoc templates were created for each phantom to enable the quantitative evaluation of
    the user accuracy. The template was designed to exactly match
    the phantom surface and it was equipped with attachment holes
    engaging into cylinders mounted on the phantoms. This features ensures
    a unique and stable mechanical placement of the template on the phantom surface. "

Round 2

Reviewer 3 Report

for me is ok